

# The Rate of Equilibration of Viscous Aerosol Particles

S. O'Meara[1], D.O. Topping[1,2], G. McFiggans[1]

[1]Centre for Atmospheric Science, School of Earth, Atmospheric & Environmental Sciences, University of Manchester, Manchester, M13 9PL, United Kingdom
[2]National Centre for Atmospheric Science (NCAS), University of Manchester, Manchester, M13 9PL, United Kingdom

*Correspondence to*: G. McFiggans (g.mcfiggans@manchester.ac.uk)

**Abstract.** The proximity of atmospheric aerosol particles to equilibrium with their surrounding condensable vapours can substantially impact their transformations, fate and impacts and is the subject of vibrant research activity. In this study we first compare equilibration timescales estimated by three different models for diffusion through aerosol particles to assess

any sensitivity to choice of model framework. Equilibration times for diffusion coefficients with varying dependencies on composition are compared for the first time. We show that even under large changes in the saturation ratio of a semi-volatile component ($e_s$) of 1-90% predicted equilibration timescales are in agreement, including when diffusion coefficients vary with composition. For condensing water and a diffusion coefficient dependent on composition, a plasticising effect is observed, leading to a decreased estimated equilibration time with increasing final $e_s$. Above 60% final $e_s$ maximum

equilibration times of around 1 s are estimated for comparatively large particles (10 μm) containing a relatively low diffusivity component ($1 \times 10^{25}$ m$^2$s$^{-1}$ in pure form). This, as well as other results here, questions whether particle-phase diffusion can be a limiting factor in gas-particle mass transfer in the ambient atmosphere, at least for water-soluble particles. In the second part of this study, we explore sensitivities associated with the use of particle radius measurements to infer diffusion coefficient dependencies on composition using a diffusion model. Given quantified similarities between models

used in this study, our results confirm considerations that must be taken into account when designing such experiments. Although quantitative agreement of equilibration timescales between models is found, further work is necessary to determine their suitability for assessing atmospheric impacts, such as their inclusion in polydisperse aerosol simulations.

## 1 Introduction

Recent attention on the phase state of atmospheric particles has motivated questions about the means to model diffusion

through them. It had been conventionally assumed that particles possess a liquid phase state, such that timescales of diffusion were much less than their atmospheric residence times. However, several recent studies present evidence that particles can exist in an amorphous solid state (Smith et al., 2002; Murray and Bertram, 2008; Virtanen et al., 2010; Vaden et al., 2011). Viscosities for amorphous solid particles will be higher than for liquid ones, resulting in lower condensed phase diffusion coefficients and potentially limiting the rate of gas-particle partitioning for condensing or evaporating compounds.





For such particles it is important to critically assess models that attempt to predict or infer the effects of diffusion limitations in order to report findings with confidence.

Fick's first and second laws of diffusion state that the rate of transport of a given component through a given area is proportional to the concentration gradient normal to the area. The Fickian diffusion coefficient ($D_i$) is the proportionality

constant between the diffusive flux and the concentration gradient (Eq. (1)).

Recent attempts at modelling diffusion through particles have centred on Fick's second law, which in spherical coordinates is:

$$\frac{\partial c_i(r,t)}{\partial t} = \frac{1}{r^2}\frac{\partial}{\partial r}\left(r^2\frac{D_i\partial c_i(r,t)}{\partial r}\right),$$  (1)

where $C$ is the concentration of species $i$, $r$ is the radius from the particle centre and $t$ is time. Fick's second law is applied

when the concentration gradient, and therefore flux, changes with time and distance, i.e. non-steady state. This study analyses and compares three approximations of Eq. (1) used to model diffusion through particles:

i) The ETH model presented by Zobrist et al. (2011), based on the Euler forward step method;

ii) The 'kinetic multi-layer model of gas-particle interactions in aerosol and clouds' (KM-GAP) (Shiraiwa et al., 2012), based on coupled differential equations;

iii) The Fick's Second Law solved by Partial Differential Equation model (hereafter referred to as Fi-PaD), a formulation of which was used in Smith et al. (2003).

A description of each model is provided below. For a system with $D_i$ independent of composition, it has been reported that Fi-PaD and KM-GAP give very similar results (Shiraiwa et al., 2010). To our knowledge however, no detailed comparison of all three approaches, including cases of $D_i$ dependent on composition, has yet been published. Despite this, a recent study

by Lienhard et al. (2015) linked the impact of particulate viscosity on ice nucleation using a composition dependent $D_i$. A critical review of these models is intended to guide those with an interest in simulating particle evolution inside instruments, chamber experiments, and the ambient atmosphere. For non-equilibrium viscous particles, diffusivity (along with other properties such as volatility) determines the temporal evolution of particle composition and size- and number-distributions (Zaveri et al., 2014). These are key factors determining aerosol impact on climate and health, therefore the choice of

diffusion model could have far-reaching consequences (Pöschl, 2005). In addition to differences in modelled particle size and composition change, inappropriate choice of model formulation and assumptions therein could lead to differences in inferred properties, such as diffusion coefficients from single particle levitation measurements (Lienhard et al., 2014; Zobrist et al., 2011).

The numerical methods employed by all three models involve discretisation in time and space. However, subtle

differences in how they define concentration gradients may induce variations in estimated diffusion rate. Therefore, it is expected that any differences in rate will increase with greater heterogeneity in the concentration-radius profile, i.e. an increasingly steep diffusion front. Such fronts have been observed when water and glassy organics diffuse through one another (Nowakowski et al., 2015). It is currently unclear which of the numerical methods investigated here, if any, is





suitable to such a situation, given the paucity of experimental data available. Indeed, the Fickian framework may not be appropriate for some systems; in polymer studies it is well known that non-Fickian diffusion occurs for many examples of liquids diffusing through glassy polymers (Thomas and Windle, 1982; Kee et al., 2005), and in such systems a narrow diffusion front is often observed (e.g. Thomas and Windle, 1982). Alternative models have been proposed, such as the free

volume model (He et al., 2006; Price et al., 2014) and the Maxwell-Stefan model (Krishna and Wesselingh, 1997). The aim of this study, however, is to compare the estimated equilibration timescales of the Fickian diffusion models that are used in atmospheric aerosol science and, in turn, assess sensitivities of derived diffusion coefficients in such particles.

## 2 Method

### 2.1 Model Description

The ETH model, KM-GAP and Fi-PaD used the same representation of an aerosol particle: it was assumed spherical, and split into concentric shells. A comparatively thin surface shell was assumed to equilibrate instantly with the gas-phase in all simulations (for the purpose of comparing particle-phase diffusion models and gaining insight into the limitation imposed by particle-phase diffusion on mass-transfer this assumption is reasonable). The initial concentration profile in bulk shells (those below the surface) was homogeneous, and in equilibrium with the initial gas phase concentration. Figure 1a

demonstrates how the particle is represented in a 2-D view. Fig. 1b illustrates water concentration-radius profiles at several time steps using the ETH model in the case of an instantaneous increase in relative humidity from 10 to 20%.

The $e$-folding time for the difference in concentration of the semi-volatile component at the surface and of its average concentration across the particle bulk was used as a metric for diffusion time by Zaveri et al. (2014). It is readily transferable to other studies, and is the chosen metric for diffusion timescale here. The ratio of the concentration difference

in the surface and bulk-average of the semi-volatile component at any time to that difference at $t = 0$ is:

$$Q = \frac{|[sv]_{eq} - \overline{[sv]}_{b,t \geq 0}|}{|[sv]_{eq} - \overline{[sv]}_{b,t=0}|}. \tag{2}$$

The $e$-folding time was therefore the time taken for $Q$ to increase/decrease by a factor of $e^1$. Figure 1b therefore demonstrates the concentration-radius profiles at several steps between $Q = 1$, which occurs at $t = 0$, and $Q = e^{-1}$, which marks the $e$-folding time. All models were run on a 12 core Intel Core i7 processor with a speed of 3.2 GHz.

Fick's first law, which assumes a constant concentration gradient and therefore flux, with time, is the basis of the ETH model. However, following the Euler forward step method, if time steps are sufficiently short this model should be able to capture changes in concentration gradient, and therefore flux, and therefore replicate the second law. The flux between shells is thus found by:

$$J_{bk,bk-1,i} = -A_{bk} D_{bk,bk-1,i} \frac{([i]_{bk} - [i]_{bk-1})}{0.5(\delta_{bk} + \delta_{bk-1})}, \tag{3}$$

where $J$ is the flux (mol s$^{-1}$) between bulk ($b$) shell numbers $k$ and $k - 1$ (ascending from $k = 1$ for the near-surface shell to $k = n$ at the centre). If $k = 1$, then $k - 1$ is the surface layer ($s$). $A$ is the surface area of the shell's outer surface.



$D_{bk,bk-1,i}$ is the diffusion coefficient at the shell boundary and here is found using one of the dependencies on composition given in Sect. 2.3. $[i]$ is concentration of compound $i$ at the shell centre and $\delta$ is shell width. Figure 1a demonstrates how these terms relate to the physical representation of the particle. From Eq. (3) the change in number of moles in a shell is found by:

$$\Delta N_{bk,i} = (J_{bk,bk-1,i} - J_{bk+1,bk,i})\Delta t, \tag{4}$$

where $\Delta t$ is the time interval (setting of $\Delta t$ is described in Sect. 2.2). The version of the ETH model we have written was tested against the model output in Zobrist et al. (2011), and found to replicate their results accurately (Fig. A1 of the appendix for the replica plot).

In KM-GAP the number of moles of a component in a shell is found by integrating the following coupled ordinary differential equations (ode) with respect to time:

$$\frac{dN_{s,i}}{dt} = (J_{b1,s,i} - J_{s,b1,i}), \tag{5}$$

$$\frac{dN_{b1,i}}{dt} = (J_{s,b1,i} - J_{b1,s,i}) + (J_{b2,b1,i} - J_{b1,b2,i}), \tag{6}$$

$$\frac{dN_{bk,i}}{dt} = (J_{bk+1,bk,i} - J_{bk,bk+1,i}) + (J_{bk-1,bk,i} - J_{bk,bk-1,i}), (k = 2, \ldots, n-1), \tag{7}$$

$$\frac{dN_{bn,i}}{dt} = (J_{bn-1,bn,i} - J_{bn,bn-1,i}). \tag{8}$$

The flux is found by:

$$J_{bk,bk\pm1,i} = K_{bk,bk\pm1,i}[i]_{bk}A_{bk}, \tag{9}$$

where $K$ is the transport rate coefficient (m s$^{-1}$):

$$K_{bk,bk\pm1,i} = \frac{2D_{bk,bk\pm1,i}}{(\delta_{bk}+\delta_{bk\pm1})}, \tag{10}$$

where $\delta$ is shell width. $D_{bk,bk\pm1,i}$ is the diffusion coefficient at the shell boundary (Sect. 2.3). Note that in Eqs. (9) and (10), if $k = 1$, then $k - 1$ is the surface shell ($s$).

Equations (5)-(10) were solved in Matlab software using the ode23tb numerical solver, which has an adaptive time step. It was found that the solver became increasingly unstable as the gradient of $D_i$ with $r$ increased, thus error tolerances (given in the appendix) were increased appropriately. ode23tb uses a Runge-Kutta method of two stages: a trapezoidal rule followed by a backward differentiation formula stage.

Fi-PaD treats Eq. (1) as an initial-boundary problem, with initial conditions:

$$C_{bk,i}(r < R_p, 0) = C_{i,eq^0}, \tag{11}$$

$$C_{s,i}(R_p, 0) = C_{i,eq}, \tag{12}$$

where $eq$ represents the equilibrium condition. Equation (11) states that initial concentrations in the bulk shells are in equilibrium with the original $e_s$ value, whilst Eq. (12) states that the initial concentration at the surface is in equilibrium with the new $e_s$. The boundary conditions were:

$$\frac{\partial N_i(0,t)}{\partial t} = 0, \tag{13}$$





$$\frac{\partial c_i(R_p,t)}{\partial t} = 0, \tag{14}$$

where $R_p$ is the particle radius. Equation (13) states that there is no flux at the centre of the particle and Eq. (14) states that the concentration of components at the surface is constant. For Fi-PaD the numerical solver pdepe in Matlab software was used. The solver uses the method of lines, which discretises the problem in space to gain a system of ordinary differential

equations that are then solved using the numerical solver ode15s in Matlab. ode15s is similar to ode23tb in that both are designed for stiff systems, however, ode15s has a high order of accuracy for a given error tolerance. The default error tolerances for pdepe were found to provide stable solutions across the range of parameter spaces used here; the contrast to the variable error tolerances used in KM-GAP is attributed to the difference in the accuracy of their ode solvers.

### 2.2 Particle Representation

Particles were assumed to consist of two components: a non-volatile ($nv$) and semi-volatile ($sv$), which were assigned the molar mass and density of sucrose and water, respectively. In general, components with relatively high molar masses are expected to have comparatively low diffusion coefficients (Haynes, 2015). To test the effect of using a high molar mass component against using sucrose on equilibration times, a molar mass (M) of 700 g mol$^{-1}$ (M of sucrose = 342.296 g mol$^{-1}$) and density ($p$) of 2.0 x10$^3$ kg m$^{-3}$ ($p$ of sucrose = 1.5805x10$^3$ kg m$^{-3}$) was assigned to the non-volatile component and its

self-diffusion coefficient was set relatively low: 1.0x10$^{-25}$ m$^2$ s$^{-1}$. When the saturation ratio of the semi-volatile component ($e_s$) increased from 1-90% the $e$-folding times for all three models increased between 13-16% from those using sucrose values (since the molar volume of the non-volatile component increased a decreased semi-volatile component concentration was required to attain equilibrium, leading to a decreased concentration gradient for the same change in $e_s$). Since these changes to $e$-folding times are similar across the models and are for a comparatively large change in $e_s$, our conclusions are

expected to be applicable to a broad range of component M and $p$ values.

All models assumed ideality for most simulations (see later) so that at equilibrium the value of $e_s$ equalled the mole fraction of the semi-volatile component in the condensed phase. While estimates of accommodation coefficients for semi-volatiles cover a wide range, for the purposes of this study we have held it constant at unity (as has been found reasonable for that of water vapour on liquid water in multiple studies, e.g. Kolb et al. (2010)). Assuming ideality, the volume of a

component was equal to the product of its number of moles ($N$) and molar volume. The volume of a shell was therefore given by:

$$V_{bk} = N_{nv,bk}\left(\frac{M_{nv}}{p_{nv}}\right) + N_{sv,bk}\left(\frac{M_{sv}}{p_{sv}}\right), \tag{15}$$

where $M$ is molar mass and $p$ is density.

To accurately simulate the size change in particles resulting from condensation (growth) or evaporation (shrinkage),

at the end of each time interval, shell volumes were recalculated using the new values of $N_{i,bk}$. In KM-GAP and Fi-PaD, a maximum change to the particle radius of 0.1% was allowed per time step; if the radius change exceeded 0.1% the interval was iteratively shortened until the change was acceptable. Decreasing this maximum acceptable change did not change $e$-





folding times significantly (<2% for both KM-GAP and Fi-PaD when a maximum radius change of 0.01 % was used instead), thus it was considered sufficient to account for volume change. For the ETH model, it has been recommended that to ensure model stability, the number of moles inside any shell should not change by more than 2% over a single time interval (Zobrist et al. 2011). The same condition was used here because values below 2% did not change predicted $e$-

folding times significantly (<1% change when maximum change in number of moles was 0.01 % instead).

Bulk shells (those below the surface) were initially set to have equal widths. The surface shell represents the sorption layer, where transfer between the condensed and gas-phase occurs. Since the surface shell is contained within the initial particle diameter, the width should be sufficiently thin to not significantly affect the $e$-folding time, i.e. one must not decrease the width of bulk shells such that diffusion is accelerated. A factor of $1\times10^{-3}$ of the particle radius was chosen to

calculate the surface shell width because using lower factors resulted in no significant change to estimated $e$-folding time. During condensation the surface shell expands; however, since this shell simulates the boundary between the shell and the gas-phase it should remain comparatively thin. Therefore, if the surface shell grew to double its initial width, it was reduced back to its initial width by transferring the excess volume to the near-surface shell, or, if this near-surface shell had a width greater than the total radius divided by the number of shells, the transferred material was used to make a new near-surface

shell. The concentration of components in the transferred material was the same as in the surface shell (i.e. at equilibrium with the gas phase). This approach had potential to introduce numerical diffusion by decreasing the distance for diffusion in the case of introducing a new shell and decreasing the concentration gradient in the case of transfer.

To gain an indication of whether numerical diffusion influenced one model more than another, $e$-folding times were found with this approach (transfer on) and without it (transfer off). For the latter case the surface shell was allowed to grow

without adjustment, leading to an unrealistically wide shell and comparatively longer equilibration times, but eliminating the possibility of numerical diffusion. If numerical diffusion affected one model more than another we would expect the difference in $e$-folding times between the transfer on and transfer off cases to vary between them. However, there was no substantial difference between models: for a change in $e_s$ of 1-90 % all models had an increase of 20-30 % in $e$-folding times from the transfer on to the transfer off case; and for a change in $e_s$ of 60-80% the increase was between 6-10%. These

differences are negligible in comparison to the several orders of magnitude change in $e$-folding times seen across the range of non-volatile component diffusivity used below.

During evaporation the width of the surface shell decreased and the mass of non-volatile component in the surface shell tended toward zero. If the surface shell decreased below a factor of $1\times10^{-1}$ of its initial width it was returned to its initial width by transferring a sufficient volume from the near-surface shell. The concentration of components in the

transferred material was equal to that in the surface shell, thus the concentration in the surface shell was maintained and any excess semi-volatile component was presumed to evaporate. Similarly, if the near-surface shell shrank to below a factor of $1\times10^{-1}$ of the initial width of the surface shell, then the two shells were coalesced into a new surface shell at equilibrium concentration. It was found that decreasing the width at which transfer and coalescence were invoked led to a decrease and convergence of predicted $e$-folding times, indicating decreasing numerical diffusion (which could occur due to steepening of





the concentration gradient through either coalescence or transfer). A decrease of no more than 1% was seen across models and changes in $e_s$ when using lower factors than $1\times10^{-1}$ of the initial width of the surface shell, thus this factor was concluded to be sufficiently low to effectively prevent numerical diffusion.

### 2.3 $D$ Dependence

At any point in the particle the diffusion coefficient of both components was the same, i.e., we assumed symmetrical diffusion coefficients, which is valid for an ideal binary mixture (Wesselingh and Bollen, 1997). We compared models using three functions of $D_i$:

i) $D_i$ independent of the semi-volatile mole fraction ($x_{sv}$) and therefore fixed throughout the simulation;

ii) $D_i$ with a logarithmic dependence on semi-volatile mole fraction (Vignes 1966):

$$D_i(x_{sv}) = D_{sv}^{0\ x_{sv}} D_{nv}^{0\ (1-x_{sv})}, \tag{16}$$

where $D_{sv}^0$ is the self-diffusion coefficient of the semi-volatile component and $D_{nv}^0$ is the self-diffusion coefficient of the non-volatile component;

iii) $D_i$ with a sigmoidal dependence on $x_{sv}$ (Lienhard et al., 2014):

$$D_i(x_{sv}) = D_{sv}^{0\ x_{sv}\propto} D_{nv}^{0\ (1-x_{sv}\propto)}, \tag{17}$$

where $\propto$ is a correction parameter given by:

$$\ln(\propto) = (1-x_w)^2[C + 3D - 4D(1-x_{sv})]. \tag{18}$$

Where the values of $C$ and $D$ were chosen as -3.105 and 3.300 respectively. Examples of these dependencies are shown in Figure 2.

For the latter two cases $D_i(x_{sv})$ was calculated within the numerical solvers of KM-GAP and Fi-PaD, whilst for the

ETH model it was calculated at the start of each time step. $x_{sv}$ at a shell boundary was found using the arithmetic mean concentration of the semi-volatile component across the bounding shells.

In the first part of the study we compare the equilibration timescales estimated by models when the diffusion coefficient is constant and when it follows the logarithmic and sigmoidal dependencies on composition given above. Self-diffusion coefficients of the non-volatile and semi-volatile components range between that of water at room temperature,

$2.0\times10^{-9}$ m$^2$s$^{-1}$ (Starr et al., 1999), and a comparatively low value of $1.0\times10^{-25}$ m$^2$s$^{-1}$, which according to the Stokes-Einstein relationship between diffusivity and viscosity, is representative of a glassy material (Debenedetti and Stillinger, 2001). $e$-folding times are found for several changes in the vapour-phase saturation ratio of a semi-volatile component, and across a range of particle sizes and differences in the self-diffusion coefficient of components.

Finally, we present an example of the differences in modelled particle size change with time when different

dependencies of $D_i$ on composition are assumed, thereby providing guidance on the most effective experimental procedure for inference of diffusion coefficient dependencies on composition. For actual inferences one would preferably have good



knowledge of the system's deviation from ideality, therefore we use the same system already presented in Zobrist et al. (2011), which includes an attempt to account for non-ideality.

## 3 Results

Numerical convergence of $e$-folding times was observed with increasing spatial resolution for all three models due to improved resolution of concentration gradients and therefore changes in $D_i$ (when dependent on composition) and flux with space. $e$-folding times showed an exponential relationship with shell number (e.g. Fig. A2 of the appendix), thus the criteria for shell number was that at which the $e$-folding time was within 10% of the asymptote. Generally as the gradient of $D_i$ with particle radius increased, the shell number increased to maintain convergence (Table A2 of the appendix). However, increasing the shell number increases the possibility of accumulating significant round-off error, in addition to requiring greater computer time. The round-off error at the chosen resolution was investigated by halving the number of significant numbers assigned to variable values. The difference in predicted $e$-folding times between the two precisions was found to be negligible, with a maximum of 2%, indicating that round-off error was not a substantial source of inaccuracy.

Zobrist et al. (2011) reported requiring up to several thousand shells in the ETH model to resolve concentration gradients. However, we found that using of the order of hundreds gave convergence for the cases with steepest concentration gradients (Fig. A2 of the appendix). The difference in required shell resolution between the studies could be due to differences in $D_i$ dependence on composition.

In the first model comparison, $e$-folding times were found when $D_i$ was independent of $x_{sv}$. For a complete analysis of model output, initial particle sizes were varied between $1\times10^{-5}$ m and $1\times10^{-8}$ m, which covers most of the size range observed in the ambient atmosphere (Seinfeld and Pandis, 2006) and $D_i$ ranged between $2.0\times10^{-9}$ m$^2$s$^{-1}$ and $1.0\times10^{-25}$ m$^2$s$^{-1}$. $e$-folding times were found across this parameter space for a change in $e_s$ of 1-90% and 90-1% for all three models. This relatively large change in $e_s$ was chosen to create a large concentration gradient, as this would most likely induce disagreement between models. However, all models agreed very well across the whole range of particle size and $D_i$ (Fig. A3).

In the next case $D_i$ varied logarithmically with mole fraction of the semi-volatile, between a maximum of $2.0\times10^{-9}$ m$^2$s$^{-1}$ at $x_{sv} = 1$ and a minimum given by $D_{nv}^0$ (i.e. $D_i$ at $x_{sv} = 0$). $D_{nv}^0$ ranged between $2.0\times10^{-9}$ and $1.0\times10^{-25}$ m$^2$s$^{-1}$. Contour plots of $e$-folding times as a function of $D_{nv}^0$ and $D_{p,t=0}$ and a 1-90% and a 90-1% change in $e_s$ are shown in Figs. 3a and 3b, respectively.

For both changes in $e_s$ there is good agreement of $e$-folding times between all models, with a maximum variation of 10 %, which is well within the uncertainty caused by varying degrees of numerical convergence and potential numerical diffusion. Diffusion times are much shorter than in the constant $D_i$ case due to the high diffusivity of the semi-volatile component. Fig. 3a shows that even when starting with a glassy particle, if the saturation ratio of a plasticising semi-volatile component increases sufficiently, the $e$-folding state can be reached in less than 1 s. For the decreasing $e_s$ used in Fig. 3b a





low diffusivity outer casing will form on the particle, impeding diffusion and evaporation. However, Fig. 3b shows that if a particle initially of water-like diffusivity is quickly dried, the $e$-folding state is reached within 10 s, even when the non-volatile component has a relatively low diffusivity.

$e$-folding times for 1-90% and 90-1% changes in $e_s$ were also found using the sigmoidal dependence of $D_i$ on $x_{sv}$;

the results are given in Figs. 3c and 3d, respectively. In the 90-1% case an unpractical computer time (>12 hours) was required to attain numerical convergence at low values of $D_{nv}^0$, therefore the minimum $D_{nv}^0$ is $1 \times 10^{-20}$ m$^2$s$^{-1}$. For this relatively large change in $e_s$ the sigmoidal dependence induces a steeper diffusion front than the logarithmic dependence. Despite this, the models show good agreement here also. In the 1-90% case, a maximum variation in $e$-folding times of 5% is seen while for 90-1% this value is 30 %. This latter variation is between KM-GAP and the other two models and is

greater than expected from different degrees of numerical convergence. However, given the gradual divergence of the $e$-folding isolines in Fig. 3d, we do not attribute the discrepancy to model framework differences, but to an insufficient shell resolution in KM-GAP. Diffusion is quicker using the sigmoidal dependence than the logarithmic dependence, particularly for the 90-1% scenario. This is explained by the higher $D_i$ values at $x_{sv}$>0.5 (Fig. 2).

$e_s$ changes more realistic of the atmosphere were also tested. Results for 60-80% and 80-60% (Fig. A4) are similar

to those for 1-90% and 90-1% for their respective $D_i$ dependency; there is good model agreement, and across the D$_{p,t=0}$ and $D_{nv}^0$ range and for both dependencies $e$-folding time is less than 1 s. Results for 10-20% and 20-10%, given in Fig. 4, also show agreement between models. For both dependencies diffusion is much slower than in the 1-90% and 60-80% simulations, approaching 1 ky at low $D_{nv}^0$ and high D$_{p,t=0}$. This shows that at low saturation ratios of semi-volatile component, gas-particle partitioning can be limited by condensed-phase diffusion in viscous particles.

$e$-folding times between models were also found to be in good agreement for these changes in $e_s$ when $D_{nv}^0$ was fixed at $1.0 \times 10^{-25}$ m$^2$s$^{-1}$ and $D_{sv}^0$ was varied between $1.0 \times 10^{-25}$ - $2 \times 10^{-9}$ m$^2$s$^{-1}$.

In the final part of this study the estimated temporal profile of particle radius was compared between the sigmoidal and logarithmic $D_i$ dependencies. We have used the non-ideality described in Zobrist et al. (2011) and the ETH model, though the results above indicate that KM-GAP and Fi-PaD would produce identical profiles. For the inference of $D_i$

dependency from radius measurements the signal to noise ratio is minimised by inducing a large change in radius relative to the measurement accuracy over a time-span that is large compared to the measurement frequency.

Taking the case of water as the semi-volatile component, from Fig. 3 it is clear that for certain values of $D_{nv}^0$ and certain changes in $e_s$ attaining a large ratio of equilibrium time to measurement frequency may be difficult, even if the change in radius is large. Indeed, the radius-time profiles in Figs. 5 and 6 for instantaneous changes in $e_s$ and a $D_{nv}^0 = 1 \times 10^{-}$

$^{25}$ m$^2$s$^{-1}$ confirm that for changes with a high final $e_s$, significant radius change is estimated to occur over less than 1 s, while the measurement frequency reported in the studies of Zobrist et al. (2011) and Lienhard et al. (2014) is approximately 15 s. Nevertheless, for the $e_s$ change of 1-90% in Fig. 5, there is a notable difference in the radius profiles between the dependencies. Despite having lower $D_i$ at low $x_{sv}$, the radius change from the sigmoidal dependence is more rapid than the





logarithmic, indicating that the $D_i$ at higher $x_{sv}$ has a dominating effect on the profile. The inference of $D_i$ dependency using such a large change in $e_s$ is therefore poorly constrained for lower $x_{sv}$. For better constraint smaller changes in $e_s$ are required, such as those used in Lienhard et al. (2014). An example of the radius profiles following incremental changes in $e_s$, $D_{nv}^0 = 1 \times 10^{-25}$ m$^2$s$^{-1}$, $D_{sv}^0 = 2 \times 10^{-9}$ m$^2$s$^{-1}$ and using both dependencies is shown in Fig. 6. This plot demonstrates the need

for consideration of the time a given $e_s$ is maintained in measurement experiments, since the difference in the equilibrium timescales between the $e_s$ increments covers orders of magnitude. Indeed, over low changes in $e_s$ such as between 1-10%, equilibration time may be too long to be practical for gaining a useful measurement of radius change. It is worthwhile to note that the rate of change of $e_s$ over an increment is preferably much greater than the rate of equilibration, as this provides the greatest potential for a clear signature of the $D_i$ dependence and therefore greatest constraint on inference.

**4 Discussion and Conclusion**

The results above show that despite variations in their numerical methods, all three Fickian-based diffusion models tested here: the ETH model, KM-GAP and Fi-PaD give good agreement of estimated $e$-folding timescales over a wide range of changes to the saturation ratio of the semi-volatile component and over a wide range of differences in the self-diffusion coefficient of the semi-volatile and non-volatile components. Furthermore, there is good agreement between models when

different dependencies of diffusion coefficient on composition are used. This result has not been reported before to our knowledge and verifies consistency between existing Fickian diffusion models. The maximum disagreement in $e$-folding times for results gained with satisfactory shell resolution is 10%, which is within the uncertainty generated by varying degrees of numerical convergence and potential numerical diffusion.

The $e$-folding times given in Fig. 3 for changes in $e_s$ of 1-90% and 90-1%, and in Fig. A4 for changes of 60-80%

and 80-60%, show that for a semi-volatile component with water-like (at room temperature) diffusivity, given a sufficiently high starting/finishing $e_s$, attainment of the $e$-folding state is effectively instant compared to residence times in the atmosphere and chamber experiments. This is due to the plasticising effect of water (and applies to any semi-volatile component with a sufficiently high self-diffusion coefficient). At lower values of $e_s$ diffusion time can be much longer (Fig. 4)), consistent with measurement studies (e.g. Zobrist et al., 2011 and Lienhard et al., 2014). The question therefore arises

that for a given $D_{sv}^0$, at what $e_s$ can equilibration be assumed instant? Figure 6 indicates that for water condensing at room temperature equilibration time is less than 1 s when the final $e_s$ is greater than 50% for the sigmoidal dependence used here and when it is greater than 60% for the logarithmic dependence. These results indicate no limitation on mass transfer of water from particle-phase diffusion at high relative humidity and at ambient temperature, and therefore no impediment to the formation of cloud droplets. Experimental results from Lienhard et al. (2014) and Zobrist et al. (2011) indicate that this is

also true down to ~250 K.

For a hygroscopicity tandem differential mobility analyser (HTDMA), which has a typical residence time of 20-25 s (Zardini et al. 2008) and $e_s$ change of <10-~90%, Figs. 3 and 5 show that equilibration is attained even when sampling





relatively large ($\sim 1 \times 10^{-5}$ m) particles containing components of relatively low diffusivity (self-diffusion coefficients $\sim 1 \times 10^{-25}$ $m^2 s^{-1}$) .

Note, however, that we have not considered extreme dependencies of $D_i$ on composition. If, for example, a very high mole fraction of water were required before the "cliff-edge" in the sigmoidal dependence (Fig. 2)) was reached, longer

diffusion times than those shown here would be expected.

Regarding the inference of diffusion coefficient dependence on composition from particle radius measurements, we have shown that incremental changes in $e_s$ provide the best constraint, and note that changes should occur over a short time compared to the equilibration time. The consistency between the diffusion models shown here indicates that the choice of model does not affect the accuracy of the inferred dependence (as long as sufficient spatial and temporal resolution is used).

In a follow-up study we intend to investigate the implementation of composition-dependent $D_i$ in the Model for Simulating Aerosol Interactions and Chemistry (MOSAIC) (Zaveri et al., 2008). MOSAIC is used for chamber and ambient studies and can therefore include, among other factors, multiple components, chemistry and volatility. Furthermore, it can model polydisperse aerosol, providing insight into how composition-dependent diffusion coefficients affect the evolution of size distributions.

As mentioned, Fickian-diffusion is, strictly speaking, limited to ideal-systems. Thus, for cases where dissolution occurs, for example, the employed or derived diffusion coefficients are actually effective values of $D_i$. As we mention briefly in the introduction, numerous alternative theories to Fickian diffusion exist. Although an analysis of such frameworks is beyond the scope of this study, a similar critical analysis may be useful in the future when data from more complex multicomponent systems exist.

**Appendix A**

To validate our version of the ETH model Figure 3d of Zobrist et al. (2011) was reproduced using our version of the model and the relative humidity measurements presented in their Fig. 3a. Note that in reproducing this figure the dependence of diffusion coefficient on water activity given in Zobrist et al. (2011) was used. Furthermore, non-ideality, as described in Zobrist et al. (2011) was used. Our reproduction is given in Fig. A1.

Figure A2 shows the convergence of $e$-folding times with increasing shell number for the ETH model. Results are for self-diffusion coefficients of the semi-volatile and non-volatile components of $2.0 \times 10^{-9}$ $m^2 s^{-1}$ and $1.0 \times 10^{-25}$ $m^2 s^{-1}$, respectively, with a logarithmic dependence of $D_i$ on composition and change to the vapour-phase saturation ratio of the semi-volatile of 1-90% (Fig. A2a) and 90-1% (Fig. A2b). These cases were chosen because they are expected to have the strongest concentration and diffusion coefficient gradients through the particle (compared to other cases in this study) and

should therefore require greatest spatial resolution. The exponential fits in Fig. A2 were obtained using Igor Pro software.

The absolute tolerances that were required to attain stability in the KM-GAP model are given in Table A1. The tolerance was dependent on the self-diffusion coefficient of the non-volatile and the initial particle diameter. The relative





tolerance was kept fixed at $1.0 \times 10^{-12}$. These tolerances were used for changes to the semi-volatile saturation ratio of 1-90% and 90-1%. Since these represent the largest changes in saturation ratio used in this study, the tolerances in Table A1 are conservative values for all other cases presented in the study.

5 The number of shells used for each model is given in Table A2. The optimum number of shells required for acceptable numerical convergence was found to be dependent on the change in semi-volatile saturation ratio and the difference in the self-diffusion coefficients of the components.

 The results for model estimates of $e$-folding times when the diffusion coefficient was kept constant are given in Fig. A3. Good agreement can be seen between all three models across the values of constant diffusion coefficient and initial particle diameter. In Fig. A4 are the $e$-folding times for changes of 60-80 and 80-60% in the saturation ratio of the semi-

10 volatile component using the logarithmic and sigmoidal dependencies of diffusion coefficient on semi-volatile component mole fraction.

**Author Contributions**

S.O. wrote the model codes, ran the simulations, created plots and tables and wrote the manuscript. D.O.T. and G.M. provided substantial input to the method, model development and manuscript.

15 **Acknowledgments**

The Natural Environment Research Council has funded this work through the Ph.D. studentship of S. O'Meara, grant number NE/K500859/1, and through grant number NE/J02175X/1.




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





**Table A1.** Absolute tolerances used in KM-GAP whilst the self-diffusion coefficient of the semi-volatile component was held constant at $2.0 \times 10^{-9}$ m$^2$s$^{-1}$ and the saturation ratio of the semi-volatile component was increased and decreased from 1-90% and 90-1%. The absolute tolerance required for stability depended on the self-diffusion coefficient of the non-volatile component ($D_{nv}^0$) and the initial particle diameter ($D_{p,t=0}$).

| $D_{nv}^0$ (m$^2$s$^{-1}$) | $D_{p,t=0}$ (m) | | | |
|---|---|---|---|---|
| | $1.0 \times 10^{-5}$ | $1.0 \times 10^{-6}$ | $1.0 \times 10^{-7}$ | $1.0 \times 10^{-8}$ |
| $1.0 \times 10^{-8}$-$1.0 \times 10^{-14}$ | $1.0 \times 10^{-12}$ | $1.0 \times 10^{-13}$ | $1.0 \times 10^{-14}$ | $1.0 \times 10^{-15}$ |
| $1.0 \times 10^{-16}$ | $1.0 \times 10^{-13}$ | $1.0 \times 10^{-14}$ | $1.0 \times 10^{-15}$ | $1.0 \times 10^{-16}$ |
| $1.0 \times 10^{-18}$ | $1.0 \times 10^{-14}$ | $1.0 \times 10^{-15}$ | $1.0 \times 10^{-16}$ | $1.0 \times 10^{-17}$ |
| $1.0 \times 10^{-20}$ | $1.0 \times 10^{-15}$ | $1.0 \times 10^{-16}$ | $1.0 \times 10^{-17}$ | $1.0 \times 10^{-18}$ |
| $1.0 \times 10^{-22}$ | $1.0 \times 10^{-16}$ | $1.0 \times 10^{-17}$ | $1.0 \times 10^{-18}$ | $1.0 \times 10^{-19}$ |
| $1.0 \times 10^{-24}$ | $1.0 \times 10^{-17}$ | $1.0 \times 10^{-18}$ | $1.0 \times 10^{-19}$ | $1.0 \times 10^{-20}$ |
| $1.0 \times 10^{-26}$ | $1.0 \times 10^{-18}$ | $1.0 \times 10^{-19}$ | $1.0 \times 10^{-20}$ | $1.0 \times 10^{-21}$ |

**Table A2.** Number of shells used in each model for each change in the vapour-phase saturation ratio of the semi-volatile component ($\Delta e_s$) and for different values of non-volatile component self-diffusion coefficient ($D_{nv}^0$).

| $D_{nv}^0$ (m$^2$s$^{-1}$) | $\Delta e_s$ | | | | | |
|---|---|---|---|---|---|---|
| | 1-90% | 90-1% | 60-80% | 80-60% | 10-20% | 20-10% |
| | ETH model # shells | | | | | |
| $1.0 \times 10^{-8}$-$1.0 \times 10^{-14}$ | 40 | 40 | 40 | 40 | 40 | 40 |
| $1.0 \times 10^{-16}$-$1.0 \times 10^{-26}$ | 300 | 300 | 40 | 40 | 40 | 40 |
| | KM-GAP # shells | | | | | |
| $1.0 \times 10^{-8}$-$1.0 \times 10^{-12}$ | 40 | 40 | 40 | 40 | 40 | 40 |





| | | | | | |
|---|---|---|---|---|---|
| $1.0\times10^{-14}$ | 60 | 60 | 40 | 40 | 40 | 40 |
| $1.0\times10^{-16}$ | 100 | 100 | 40 | 40 | 40 | 40 |
| $1.0\times10^{-18}$ | 200 | 200 | 40 | 40 | 40 | 40 |
| $1.0\times10^{-20}$ | 250 | 250 | 100 | 100 | 40 | 40 |
| $1.0\times10^{-22}$ | 270 | 270 | 100 | 100 | 40 | 40 |
| $1.0\times10^{-24}$ | 300 | 300 | 100 | 100 | 40 | 40 |
| $1.0\times10^{-26}$ | 330 | 330 | 100 | 100 | 40 | 40 |
| Fi-PaD # shells | | | | | |
| $1.0\times10^{-8}$- $1.0\times10^{-14}$ | 40 | 40 | 40 | 40 | 40 | 40 |
| $1.0\times10^{-16}$- $1.0\times10^{-26}$ | 300 | 300 | 80 | 80 | 40 | 40 |



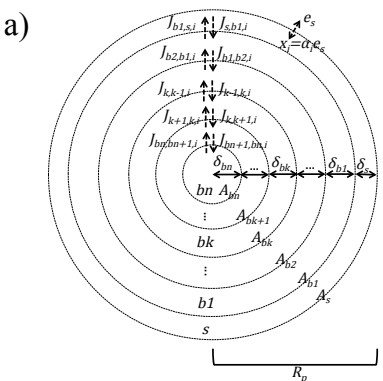

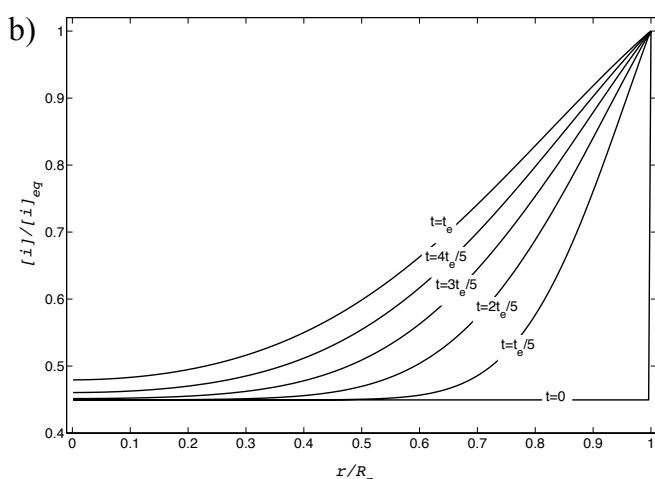

**Figure 1.** a) A schematic of a particle split into shells, as used in the diffusion models, with shell boundaries represented by dashed lines, and symbols relating to those used in the model equations. The relative width of the surface shell is shown larger than that used in models for clarity. b) The concentration-radius profiles at various times during diffusion for a semi-volatile component diffusing inwards, where $t_e$ is the $e$-folding time. Note that the axes are relative, and normalised by the total radius ($R_p$) of the particle on the abscissa and by the equilibrium concentration ($[i]_{eq}$) on the ordinate.





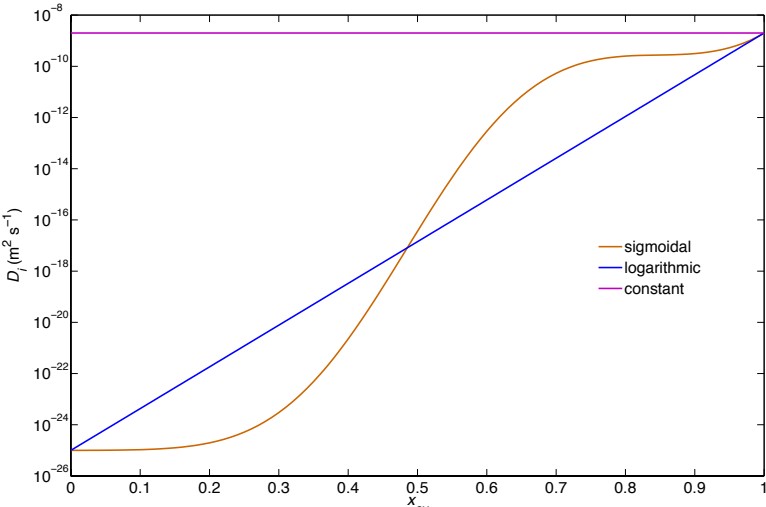

**Figure 2.** Example dependencies of $D_i$ on the mole fraction of the semi-volatile component. For the constant case both components have a value of $2\times10^{-9}$ m$^2$s$^{-1}$ and for the other cases the self-diffusion coefficient of the semi-volatile component is set to $2\times10^{-9}$ m$^2$s$^{-1}$ and that of the non-volatile is $1\times10^{-25}$ m$^2$s$^{-1}$.

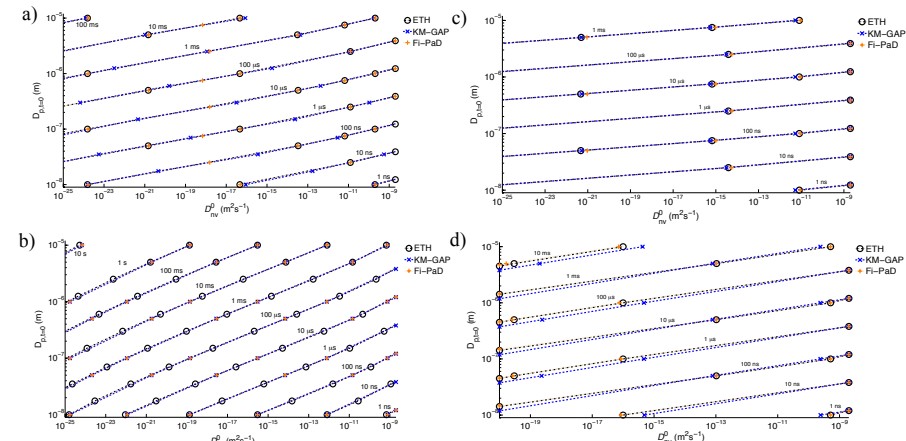

**Figure 3.** $e$-folding time contour plots for different changes to the saturation ratio of the semi-volatile component ($\Delta e_s$) and different diffusion coefficient dependencies: a) $\Delta e_s = 1$-90% logarithmic dependence, b) $\Delta e_s = 90$-1%, logarithmic



dependence, c) $\Delta e_s$ = 1-90%, sigmoidal dependence, and d) $\Delta e_s$ = 90-1% sigmoidal dependence. $D_{nv}^0$ is the diffusion coefficient at a semi-volatile mole fraction of 0, while $D_{sv}^0$ (diffusion coefficient at a semi-volatile mole fraction of 1) was fixed at 2.0x10⁻⁹ m²s⁻¹.

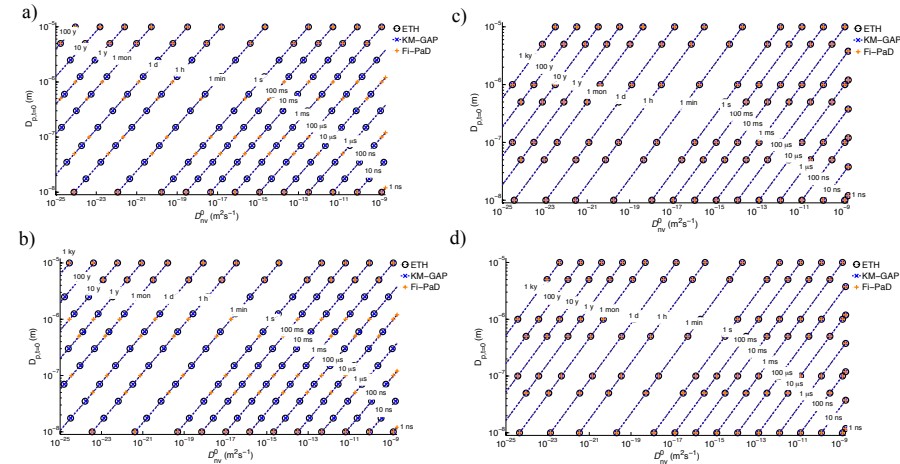

**Figure 4.** *e*-folding time contour plots for different changes to the saturation ratio of the semi-volatile component ($\Delta e_s$) and different diffusion coefficient dependencies: a) $\Delta e_s$ = 10-20% logarithmic dependence, b) $\Delta e_s$ = 20-10%, logarithmic dependence, c) $\Delta e_s$ = 10-20%, sigmoidal dependence, and d) $\Delta e_s$ = 20-10% sigmoidal dependence. $D_{nv}^0$ is the diffusion coefficient at a semi-volatile mole fraction of 0, while $D_{sv}^0$ (diffusion coefficient at a semi-volatile mole fraction of 1) was

10  fixed at 2.0x10⁻⁹ m²s⁻¹.





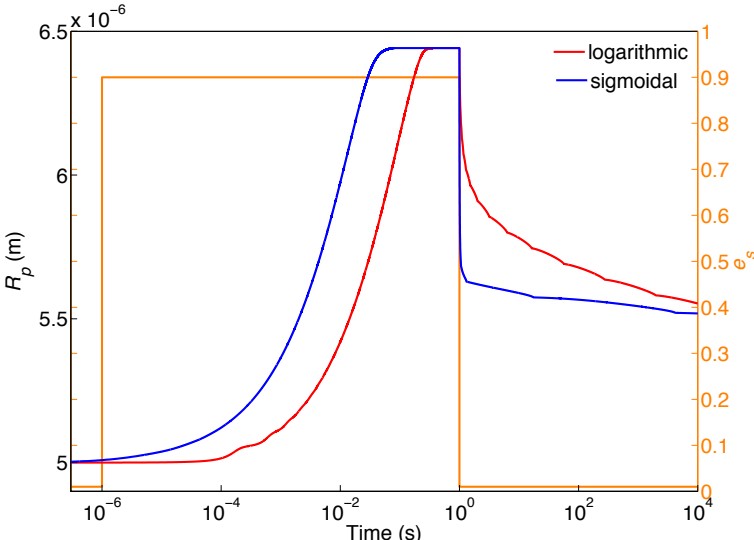

**Figure 5.** The radius ($R_p$) change with time for a single particle subject to the changes in $e_s$ shown by the orange curve and right vertical axis. Here $D_{sv}^0 = 2\text{x}10^{-9}$ m$^2$s$^{-1}$ and $D_{nv}^0 = 1\text{x}10^{-25}$ m$^2$s$^{-1}$ using the $D_i$ dependencies given in the legend.



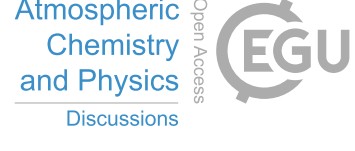

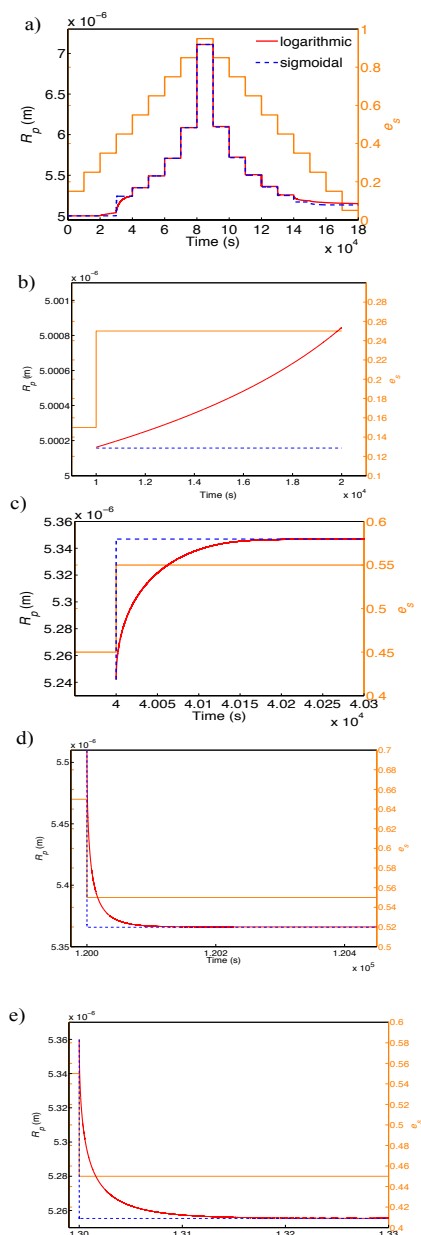





**Figure 6.** a) Radius ($R_p$) change with time for a single particle experiencing the changes in saturation ratio of the semi-volatile component ($e_s$) shown by the orange curve (allied with the right vertical axis), for $D_{sv}^0 = 2\times10^{-9}$ m$^2$s$^{-1}$ and $D_{nv}^0 = 1\times10^{-25}$ m$^2$s$^{-1}$ and using the $D_i$ dependencies given in the legend. b), c), d), e): time intervals of a) over select changes in $e_s$ (as shown by their orange curve and right vertical axes).

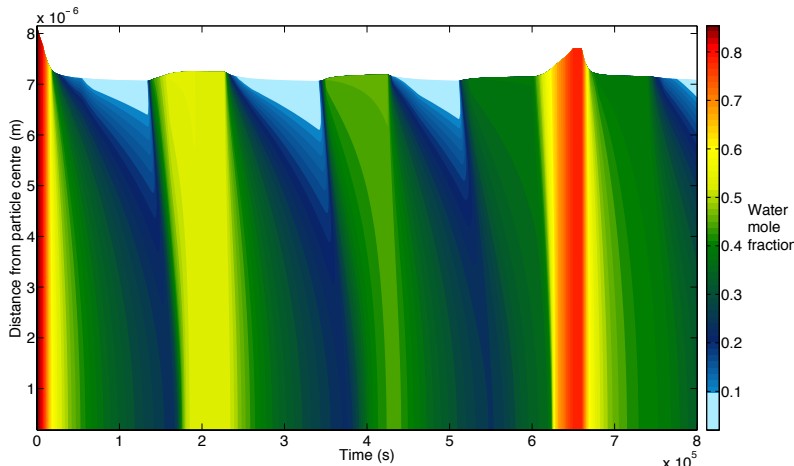

**Figure A1.** The water mole fraction as a function of time and distance through a single particle using the ETH model and the same inputs as for Fig. 3 of Zobrist et al. (2011).





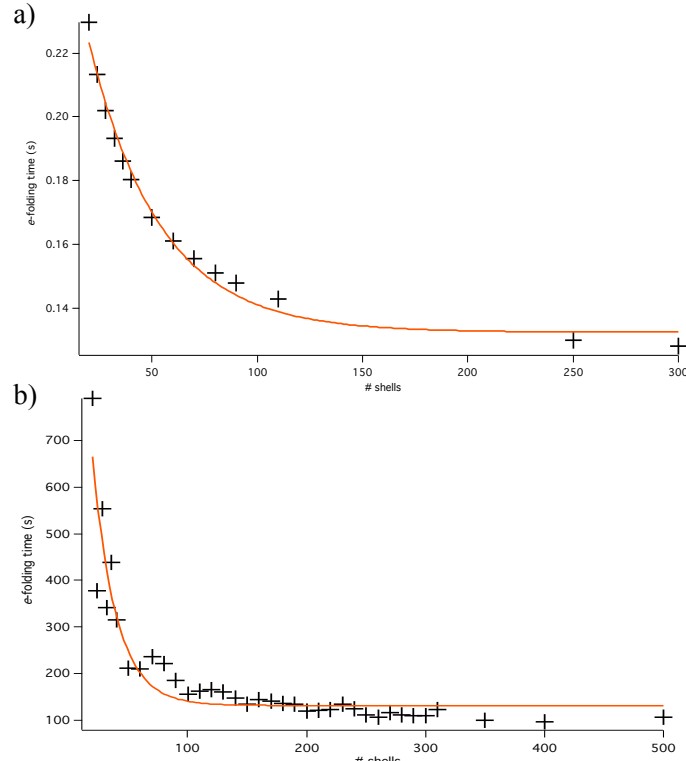

**Figure A2.** The *e*-folding time convergence with increasing shell number for the ETH model: change in the semi-volatile component saturation ratio in a) was 1-90% and in b) was 90-1%. The self-diffusion coefficient of the semi-volatile was $2.0 \times 10^{-9}$ m$^2$s$^{-1}$ and that of the non-volatile was $1.0 \times 10^{-25}$ m$^2$s$^{-1}$. Orange curves are the exponential best fits.





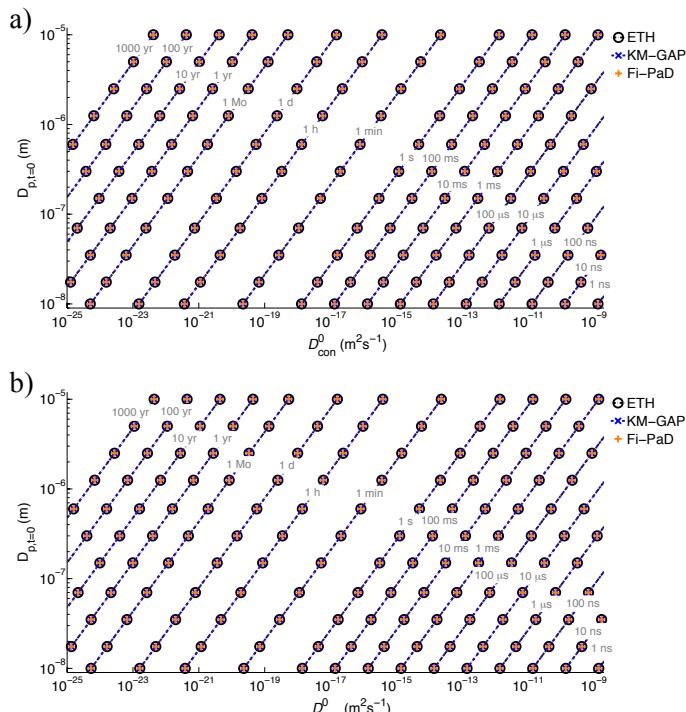

**Figure A3.** *e*-folding times (isolines) for the three models given in the legend. Where $D_{con}^0$ is the constant diffusion coefficient used throughout the simulation. $D_{p,t=0}$ is the initial particle diameter. In a) the saturation ratio of the semi-volatile is increased from 1-90% instantaneously, whilst in b) it is decreased from 90-1% instantaneously.





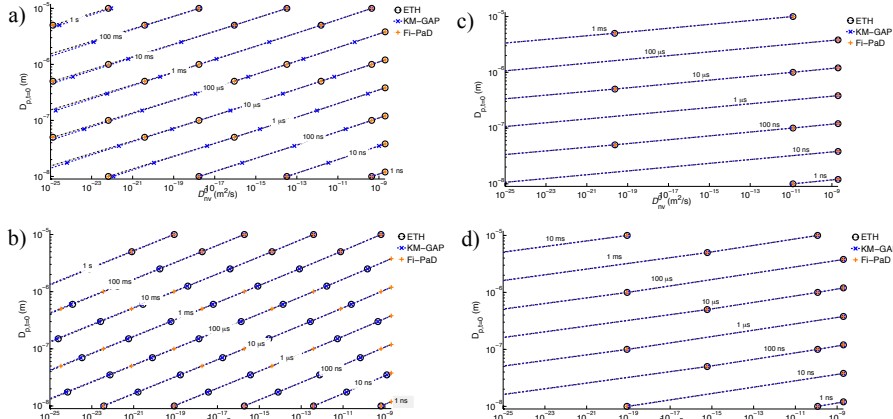

**Figure A4.** *e*-folding time contour plots for different instantaneous changes in the saturation ratio of the semi-volatile component ($\Delta e_s$) and different diffusion coefficient dependencies: a) $\Delta e_s$ = 60-80% and logarithmic dependence, b) $\Delta e_s$ = 80-60% and logarithmic dependence, c) $\Delta e_s$ = 60-80% and sigmoidal dependence, and d) $\Delta e_s$ = 80-60% and sigmoidal dependence. Models are given in the legend.