# Peer review of "The Rate of Equilibration of Viscous Aerosol Particles"

_Atmospheric Chemistry and Physics, 2015_

## Referee Comment (RC1) · Anonymous Referee #1 · 11 Feb 2016

This manuscript illustrates that three different models previously used to predict equilibrium times in particles give the same results. This manuscript also illustrates how mixing times in particles can vary as a function of diffusion coefficient, particle diameter, and change in saturation ratio of the volatile component. Finally the manuscript illustrates appropriate experimental conditions for determining diffusion coefficients from particle growth experiments. Since equilibration times in viscous aerosols is currently receiving considerable attention in the atmospheric community, this manuscript is appropriate for publication in ACP. I recommend publication after the following minor comments are addressed.

1) The modelling studies and results from this manuscript are most relevant for the hygroscopic growth of water soluble particles. This point is made clearly in the Discussion and Conclusions Section. However, I think this point should be made more clearly in

the Abstract and Introduction. For example, line 17 of the abstract "this as well as other results here questions whether particle-phase diffusion can be a limiting factor in gas-particle mass transfer in the ambient atmosphere, at least for water-soluble particles". This sentence may be clearer if they point out that they are referring to mainly water vapor-particle mass transfer. Perhaps something like the following may be clearer: "this as well as other results here question whether particle-phase diffusion can be a limiting factor in the hygroscopic growth of atmospheric particles".

2) Equation 2. For clarity, please define the variables in the equation.

3) Three different functions were used to describe diffusion coefficients. It would be useful to indicate why the logarithmic dependence and sigmoidal dependence were chosen. For example, are these dependences consistent with theory or are they consistent with experimental results for systems like water and sucrose?

4) End of section 2.3. At this location, please give more details on the system already presented by Zobrist et al. (2011), so the reader has a better idea of the type of systems the results apply to.

5) Page 9, line 23. Please give more details on what you mean by "we have used the non-ideality described in Zobrist (2011)".

6) The contour plots were too small for me to read. I suggest making bigger figures or figures with larger fonts.

---

## Referee Comment (RC2) · Anonymous Referee #2 · 12 Feb 2016

Review of manuscript acp-2015-1019 "The Rate of Equilibration of Viscous Aerosol Particles" by S. O'Meara, D. O. Topping and G. McFiggans

The authors compare three model frameworks treating condensed phase diffusion in aerosol particles taking into account a concentration dependence of the diffusion coefficient. Applying the models to measurement techniques they investigate how to design experiments to constrain measured diffusion coefficients best.

The topic of the paper is well suited for publication in ACP, the manuscript is well written and organized and the conclusions are all supported by the modeling and illustrated with appropriate figures. It is my pleasure to recommend it for publication in ACP.

The authors may consider the following comments before submitting the final version.

In the last paragraph of the introduction the authors mention that a Fickian framework may not be appropriate for some systems, in particular glassy polymers. While an

extensive discussion is clearly beyond the scope of the manuscript, it will help the reader if the authors give some basic information in a few sentences on what causes non-Fickian diffusion, i.e. mechanical deformation in response of a solvent diffusing in a matrix.

Discussion of Fig. 1b and the figure: For a reader not familiar with the topic it would be helpful to introduce first the case of Fickian diffusion with a concentration independent diffusion constant and only then compare to one in which the diffusing species acts as a plasticizer. Also, the caption of Fig. 1b should contain the info that it shows the response to an instantaneous increase in RH from 10 to 20%.

The authors use e-folding times as a metric for diffusion time. Of course this is technically correct, but may lead to a misunderstanding for a reader who is not familiar with the topic. The temporal response of system in which the diffusing species acts as a plasticizer cannot be described by a single exponential. The approach of the authors to take as a measure the time when the difference between average bulk and surface concentration has changed by e is valid, but a short discussion is appropriate.

In the same context: I would very much appreciate at least one example in which the authors do not only compare e-folding times (in the sense mentioned above), but directly compare calculated profiles with the three models for a case leading to a steep diffusion profile. Did they observe any differences between the models here?

And last: there is no discussion on how the models compare in terms of computational speed. Of course this my dependent on the specific coding and comparison may not be easy, but either there is a significant difference between the ones coded by the authors or not. Whatever is the answer, it is of interest for a reader who would like to use one of the model framework

Technical: The Lienhard et al. 2015 paper is published now in ACP.

---

## Author Comment (AC2) · 4 Apr 2016

Point 4 of reviewer 1, regarding how non-ideality was incorporated into the diffusion model was discussed in our first response, however we have made further modifications. We present these below, with the original reviewer comment given first beside the marker 4), our response given beside the markers AC) and the changes to the paper given in quotation marks.

4) End of section 2.3. At this location, please give more details on the system already presented by Zobrist et al. (2011), so the reader has a better idea of the type of systems the results apply to.

AC) pp.8 line 20 changed to be more informative of how non-ideality accounted for:

" For actual inferences one would preferably have good knowledge of the system's deviation from ideality. In an attempt to replicate a real system, we therefore use the

estimation for water activity and density as a function of sucrose weight fraction presented in Zobrist et al. (2011). The initial and surface shell water activity were set equal to the initial and current gas-phase saturation ratio of water (the saturation ratio changed with time), with the accommodation coefficient of water assumed to be one."

AC) and pp. 10 line 24 now reads:

" We have used the water activity and density dependence on sucrose weight fraction as described in Zobrist et al. (2011) for the sucrose-water system in an attempt to replicate a non-ideal system. The ETH model was employed, though the results above indicate that KM-GAP and Fi-PaD would produce identical profiles."

---

## Author Response (AR2)

We thank the reviewer for helpful comments. We have no issue with the review and have gratefully modified the paper, as described below. Author comments are numbered and italicised while our responses are given below each comment and indented.

*1) The modelling studies and results from this manuscript are most relevant for the hygroscopic growth of water soluble particles. This point is made clearly in the Discussion and Conclusions Section. However, I think this point should be made more clearly in the Abstract and Introduction. For example, line 17 of the abstract "this as well as other results here questions whether particle-phase diffusion can be a limiting factor in gas- particle mass transfer in the ambient atmosphere, at least for water-soluble particles". This sentence may be clearer if they point out that they are referring to mainly water vapor-particle mass transfer. Perhaps something like the following may be clearer: "this as well as other results here question whether particle-phase diffusion can be a limiting factor in the hygroscopic growth of atmospheric particles".*

This is a valid point, pp. 1 line 16 of the abstract and pp. 3 line 7 of the introduction have been changed to accommodate it:

" This, as well as other results here, questions whether particle-phase diffusion through water-soluble particles can limit hygroscopic growth in the ambient atmosphere"

and

" In most test cases below the diffusing semi-volatile component has the self-diffusion coefficient of water at room temperature, and the resulting diffusion timescales are most relevant to water and water-soluble particles, however, the findings regarding consistency between models are applicable to components with self-diffusion coefficients across the investigated range ($2x10^{-9}$ - $1x10^{-25}$ $m^2s^{-1}$). "

*2) Equation 2. For clarity, please define the variables in the equation.*

pp. 3 line 21 changed to explain the terms in Eq. 2:

" The *e*-folding time for the difference in concentration of the semi-volatile component at the surface ($[sv]_{eq}$) and of its average concentration across the particle bulk ($\overline{[sv]}_b$) was used as a metric for diffusion time by Zaveri et al. (2014). It is readily transferable to other studies, and is

the chosen metric for diffusion timescale here. The ratio of the concentration difference in the surface and bulk-average of the semi-volatile component at any time ($t$) to that difference at $t = 0$ is:"

3) *Three different functions were used to describe diffusion coefficients. It would be useful to indicate why the logarithmic dependence and sigmoidal dependence were chosen. For example, are these dependences consistent with theory or are they consistent with experimental results for systems like water and sucrose?*

pp. 7 line 17 was changed to give more detail on the provenance of the logarithmic dependence:

" ii) $D_i$ with a logarithmic dependence on semi-volatile mole fraction, which has been observed for ideal systems by Vignes (1966):"

pp. 7 line 22 was changed to give more detail on the provenance of the sigmoidal dependence:

"iii) $D_i$ with a sigmoidal dependence on $x_{sv}$, which was observed for the citric acid-water system by Lienhard et al. (2014):"

pp. 7 line 26 was changed to describe the advantage of using these dependencies:

"These provided a relatively steep "cliff-edge" sigmoidal dependence and therefore a substantial variation from the logarithmic dependence, enabling a test of consistency between models across a wide range of dependencies"

4) *End of section 2.3. At this location, please give more details on the system already presented by Zobrist et al. (2011), so the reader has a better idea of the type of systems the results apply to.*

pp.8 line 20 changed to be more informative of how non-ideality accounted for:

" For actual inferences one would preferably have good knowledge of the system's deviation from ideality. In an attempt to replicate a real system, we therefore use the estimation for water activity and density as a function of sucrose weight fraction presented in Zobrist et al. (2011). The initial and surface shell water activity were set equal to the initial and current gas-phase

saturation ratio of water (the saturation ratio changed with time), with the accommodation coefficient of water assumed to be one."

and pp. 10 line 24 now reads:

" We have used the water activity and density dependence on sucrose weight fraction as described in Zobrist et al. (2011) for the sucrose-water system in an attempt to replicate a non-ideal system.  The ETH model was employed, though the results above indicate that KM-GAP and Fi-PaD would produce identical profiles."

*5) Page 9, line 23. Please give more details on what you mean by "we have used the non-ideality described in Zobrist (2011)".*

More details have been provided, pp. 10, line 10 now reads:

"We have used the non-ideal dependence of density on water weight fraction as described in Zobrist et al. (2011) for the sucrose-water system and the ETH model, though the results above indicate that KM-GAP and Fi-PaD would produce identical profiles"

*6) The contour plots were too small for me to read. I suggest making bigger figures or figures with larger fonts.*

Contour plots were modified so that text was larger, e.g.

[Figure]

Author Response to Reviewer 2

We thank the reviewer for their helpful comments. All comments were thought beneficial to our study and we have described the resulting modifications to the paper below. Reviewer comment are numbered and italicised while our responses are given below each comment and indented.

*1) In the last paragraph of the introduction the authors mention that a Fickian framework may not be appropriate for some systems, in particular glassy polymers. While an extensive discussion is clearly beyond the scope of the manuscript, it will help the reader if the authors give some basic information in a few sentences on what causes non-Fickian diffusion, i.e. mechanical deformation in response of a solvent diffusing in a matrix.*

A brief description of the process leading to divergence from Fickian diffusion has been included. p.2 line 5 now reads:

" Non-Fickian diffusion results from structural changes following diffusion and the resultant composition change. It arises when the rate of deformation is comparable to that of diffusion (Crank 1975)."

*2) Discussion of Fig. 1b and the figure: For a reader not familiar with the topic it would be helpful to introduce first the case of Fickian diffusion with a concentration independent diffusion constant and only then compare to one in which the diffusing species acts as a plasticizer. Also, the caption of Fig. 1b should contain the info that it shows the response to an instantaneous increase in RH from 10 to 20%.*

A further subplot was added to Figure 1 to illustrate the differences in concentration-radius profiles for different dependencies of the diffusion coefficient on concentration. The associated discussion starting at pp. 3 line 21 now reads:

Figure 1a demonstrates how the particle is represented in a 2-D view. Fig. 1b illustrates the concentration-radius profiles of a semi-volatile component at several time steps using the ETH model in the case of an instantaneous increase in saturation ratio from 1 to 90% when the diffusion coefficient is independent of composition. In contrast, Fig. 1c shows the same information but when the diffusion coefficient has a logarithmic dependence on composition and the self-diffusion coefficients of the two components are very different, that of the non-volatile $(D_{nv}^{0})$ = $1x10^{-21}$ $m^2s^{-1}$ and for the semi-volatile $(D_{sv}^{0})$ = 2x10-9 $m^2s^{-1}$. The "diffusion front" is clear in this example and arises from the very different diffusion coefficient values in neighbouring shells that result from variations in shell composition.

*3) The authors use e-folding times as a metric for diffusion time. Of course this is technically correct, but may lead to a misunderstanding for a reader who is not familiar with the topic. The temporal response of system in which the diffusing species acts as a plasticizer cannot be described by a single exponential. The approach of the authors to take as a measure the time when the difference between average bulk and surface concentration has changed by e is valid, but a short discussion is appropriate.*

This point is welcome and has motivated extra discussion in the method, pp. 4 line 6 now reads:

"Comparing *e*-folding times between models strictly only tests model consistency at this particular stage of diffusion and not before this. However, *e*-folding time agreement would indicate agreement at

previous times (and future ones), because the underlying equations are identical. For reassurance on this, concentration-radius profiles at times prior to *e*-folding were compared."

*4) In the same context: I would very much appreciate at least one example in which the authors do not only compare e-folding times (in the sense mentioned above), but directly compare calculated profiles with the three models for a case leading to a steep diffusion profile. Did they observe any differences between the models here?*

Providing the example described benefits the evaluation of model consistency. The modifications to the paper described below are associated with the modification described in reviewer point 3). A new figure (Fig. 5) has been included to present the agreement between models for concentration-radius profiles.

pp. 10 line 15 now reads:

" As discussed, the agreement between models in estimating *e*-folding times indicates that the estimated profiles of concentration with particle radius prior to the *e*-folding state are consistent between models because the underlying equations are the same. By comparing concentration-radius profiles at various stages of diffusion we indeed found good model agreement across all cases. In Fig. 5 we show the example of the logarithmic dependence of $D_i$ on $x_{sv}$, an instantaneous change in saturation ratio of 1-90% and with $D_{nv}^0=1\times10^{-21}$ m$^2$s$^{-1}$ and $D_{sv}^0=2\times10^{-9}$ m$^2$s$^{-1}$. At several times preceding and including *e*-folding time the concentration-radius profiles are in good agreement."

and pp. 11 line 19 reads:

" The consistency in modelled concentration-radius profiles at times preceding and including the *e*-folding state (Fig. 5) shows that if used for a polydisperse aerosol population, the models would give agreement in changes to the size distribution. In addition, if the diffusing component were reactive the rate of particle-phase reaction would depend on its concentration; therefore model agreement in concentration-radius profiles would give consistent reaction rates across the particle (which in turn could affect diffusion rate)."

*5) And last: there is no discussion on how the models compare in terms of computational speed. Of course this my dependent on the specific coding and comparison may not be easy, but either there is a significant difference*

*between the ones coded by the authors or not. Whatever is the answer, it is of interest for a reader who would like to use one of the model framework*

This additional information we agree would be beneficial to readers, so we have included the following illustration of computational speed variations in pp. 9 line 4:

5     "Using the Matlab software it was found that computational time for the case of diffusion coefficient independent of composition was quickest, gradually increasing as the steepness of the diffusion coefficient dependence on composition increased, largely due to the greater spatial resolution. For $D_i$ independent of composition the ETH model took of the order 1 s to reach the *e*-folding state while KM-GAP and Fi-PaD were of the order $10^2$ s. For a steep diffusion coefficient dependence, the chosen

10     example was the logarithmic dependence, with $D_{nv}^0 = 1 \times 10^{-25}$ m$^2$s$^{-1}$ and $D_{sv}^0 = 2 \times 10^{-9}$ m$^2$s$^{-1}$ and $e_s$ instantaneously increased from 1-90%: the ETH model took of the order $10^2$ s while both KM-GAP and Fi-PaD took of the order $10^4$ s. "

Furthermore, we have a paragraph to the discussion and conclusion, pp. 11 line 25 now says:

[revised manuscript text omitted]

Simon O'Meara 15/3/16 12:04
**Comment [2]:** reviewer 2, comment 1: description of what causes non-Fickian diffusion

Simon O'Meara 15/3/16 12:06
**Comment [3]:** reviewer 1 comment 1: sentence added for greater clarity that equilibrium timescales are relevant to water

Simon O'Meara 15/3/16 12:08
**Comment [4]:** reviewer 2, comment 2: concentration-radius profile examples are provided for two extreme cases D dependence on composition

Simon O'Meara 15/3/16 12:12
**Comment [5]:** reviewer 1 comment 2): explanation of terms in eq. 2

readily transferable to other studies, and is the chosen metric for diffusion timescale here. The ratio of the concentration difference in the surface and bulk-average of the semi-volatile component at any time ($t$) to that difference at $t = 0$ is:

$$Q = \frac{|[sv]_{eq} - \overline{[sv]}_{b,t\geq0}|}{|[sv]_{eq} - \overline{[sv]}_{b,t=0}|}. \tag{2}$$

The $e$-folding time was therefore the time taken for $Q$ to increase/decrease by a factor of $e^1$. Figures 1b and 1c therefore

5    demonstrate the concentration-radius profiles at several steps between $Q = 1$, which occurs at $t = 0$, and $Q = e^{-1}$, which marks the $e$-folding time. Comparing $e$-folding times between models strictly only tests model consistency at this particular stage of diffusion and not before this. However, $e$-folding time agreement would indicate agreement at previous times (and future ones), because the underlying equations are identical. For reassurance on this, concentration-radius profiles at times prior to the $e$-folding time were compared. All models were run on a 12 core Intel Core i7 processor with a speed of 3.2

10    GHz.

      Fick's first law, which assumes a constant concentration gradient and therefore flux, with time, is the basis of the ETH model. However, following the Euler forward step method, if time steps are sufficiently short this model should be able to capture changes in concentration gradient, and therefore flux, and therefore replicate the second law. The flux between shells is thus found by:

15    $$J_{bk,bk-1,i} = -A_{bk}D_{bk,bk-1,i}\frac{([i]_{bk}-[i]_{bk-1})}{0.5(\delta_{bk}+\delta_{bk-1})}, \tag{3}$$

where $J$ is the flux (mol s$^{-1}$) between bulk ($b$) shell numbers $k$ and $k-1$ (ascending from $k = 1$ for the near-surface shell to $k = n$ at the centre). If $k = 1$, then $k-1$ is the surface layer ($s$). $A$ is the surface area of the shell's outer surface. $D_{bk,bk-1,i}$ is the diffusion coefficient at the shell boundary and here is found using one of the dependencies on composition given in Sect. 2.3. $[i]$ is concentration of component $i$ at the shell centre and $\delta$ is shell width. Figure 1a demonstrates how

20    these terms relate to the physical representation of the particle. From Eq. (3) the change in number of moles in a shell is found by:

$$\Delta N_{bk,i} = (J_{bk,bk-1,i} - J_{bk+1,bk,i})\Delta t, \tag{4}$$

where $\Delta t$ is the time interval (setting of $\Delta t$ is described in Sect. 2.2). The version of the ETH model we have written was tested against the model output in Zobrist et al. (2011), and found to replicate their results accurately (Fig. A1 of the

25    appendix for the replica plot).

      In KM-GAP the number of moles of a component in a shell is found by integrating the following coupled ordinary differential equations (ode) with respect to time:

$$\frac{dN_{s,i}}{dt} = (J_{b1,s,i} - J_{s,b1,i}), \tag{5}$$

$$\frac{dN_{b1,i}}{dt} = (J_{s,b1,i} - J_{b1,s,i}) + (J_{b2,b1,i} - J_{b1,b2,i}), \tag{6}$$

30    $$\frac{dN_{bk,i}}{dt} = (J_{bk+1,bk,i} - J_{bk,bk+1,i}) + (J_{bk-1,bk,i} - J_{bk,bk-1,i}), (k = 2,\dots,n-1), \tag{7}$$

$$\frac{dN_{bn,i}}{dt} = (J_{bn-1,bn,i} - J_{bn,bn-1,i}). \tag{8}$$

Simon O'Meara 15/3/16 12:13
**Comment [6]:** reviewer 2 comment 3: the limitation of using e-folding times for assessing model consistency

[revised manuscript text omitted]

Simon O'Meara 15/3/16 12:29
**Comment [7]:** reviewer 1 comment 3: sentence added to better describe why the dependencies were chosen, in addition to slight changes to the small intros to their equation above

Simon O'Meara 3/4/16 11:00
**Comment [8]:** reviewer 1 comment 4: greater detail on how non-ideality accounted for and for what system it's relevant

numbers assigned to variable values. The difference in predicted $e$-folding times between the two precisions was found to be negligible, with a maximum of 2%, indicating that round-off error was not a substantial source of inaccuracy.

Zobrist et al. (2011) reported requiring up to several thousand shells in the ETH model to resolve concentration gradients. However, we found that using of the order of hundreds gave convergence for the cases with steepest concentration gradients (Fig. A2 of the appendix). The difference in required shell resolution between the studies could be due to differences in $D_i$ dependence on composition. Using the Matlab software it was found that computational time for the case of diffusion coefficient independent of composition was quickest, gradually increasing as the steepness of the diffusion coefficient dependence on composition increased, largely due to the greater spatial resolution. For $D_i$ independent of composition the ETH model took of the order 1 s to reach the $e$-folding state while KM-GAP and Fi-PaD were of the order $10^2$ s. For a steep diffusion coefficient dependence, the chosen example was the logarithmic dependence, with $D_{nv}^0 = 1 \times 10^{-25}$ $m^2 s^{-1}$ and $D_{sv}^0 = 2 \times 10^{-9}$ $m^2 s^{-1}$ and $e_s$ instantaneously increased from 1-90%: to reach $e$-folding states the ETH model took of the order $10^2$ s while both KM-GAP and Fi-PaD took of the order $10^4$ s.

In the first model comparison, $e$-folding times were found when $D_i$ was independent of $x_{sv}$. For a complete analysis of model output, initial particle sizes were varied between $1 \times 10^{-5}$ m and $1 \times 10^{-8}$ m, which covers most of the size range observed in the ambient atmosphere (Seinfeld and Pandis, 2006) and $D_i$ ranged between $2.0 \times 10^{-9}$ $m^2 s^{-1}$ and $1.0 \times 10^{-25}$ $m^2 s^{-1}$. $e$-folding times were found across this parameter space for a change in $e_s$ of 1-90% and 90-1% for all three models. This relatively large change in $e_s$ was chosen to create a large concentration gradient, as this would most likely induce disagreement between models. However, all models agreed very well across the whole range of particle size and $D_i$ (Fig. A3).

In the next case $D_i$ varied logarithmically with mole fraction of the semi-volatile, between a maximum of $2.0 \times 10^{-9}$ $m^2 s^{-1}$ at $x_{sv} = 1$ and a minimum given by $D_{nv}^0$ (i.e. $D_i$ at $x_{sv} = 0$). $D_{nv}^0$ ranged between $2.0 \times 10^{-9}$ and $1.0 \times 10^{-25}$ $m^2 s^{-1}$. Contour plots of $e$-folding times as a function of $D_{nv}^0$ and $D_{p,t=0}$ and a 1-90% and a 90-1% change in $e_s$ are shown in Figs. 3a and 3b, respectively.

For both changes in $e_s$ there is good agreement of $e$-folding times between all models, with a maximum variation of 10 %, which is well within the uncertainty caused by varying degrees of numerical convergence and potential numerical diffusion. Diffusion times are much shorter than in the constant $D_i$ case due to the high diffusivity of the semi-volatile component. Fig. 3a shows that even when starting with a glassy particle, if the saturation ratio of a plasticising semi-volatile component increases sufficiently, the $e$-folding state can be reached in less than 1 s. For the decreasing $e_s$ used in Fig. 3b a low diffusivity outer casing will form on the particle, impeding diffusion and evaporation. However, Fig. 3b shows that if a particle initially of water-like diffusivity is quickly dried, the $e$-folding state is reached within 10 s, even when the non-volatile component has a relatively low diffusivity.

$e$-folding times for 1-90% and 90-1% changes in $e_s$ were also found using the sigmoidal dependence of $D_i$ on $x_{sv}$; the results are given in Figs. 3c and 3d, respectively. In the 90-1% case an unpractical computer time (>12 hours) was

Simon O'Meara 15/3/16 11:41
**Comment [9]:** reviewer 2 comment 5) - the order of magnitude for computer time taken by each model is compared for two extreme cases.

required to attain numerical convergence at low values of $D_{nv}^0$, therefore the minimum $D_{nv}^0$ is $1\times10^{-20}$ m$^2$s$^{-1}$.  For this relatively large change in $e_s$ the sigmoidal dependence induces a steeper diffusion front than the logarithmic dependence.  Despite this, the models show good agreement here also.  In the 1-90% case, a maximum variation in $e$-folding times of 5% is seen while for 90-1% this value is 30 %.  This latter variation is between KM-GAP and the other two models and is

5  greater than expected from different degrees of numerical convergence.  However, given the gradual divergence of the $e$-folding isolines in Fig. 3d, we do not attribute the discrepancy to model framework differences, but to an insufficient shell resolution in KM-GAP.  Diffusion is quicker using the sigmoidal dependence than the logarithmic dependence, particularly for the 90-1% scenario.  This is explained by the higher $D_i$ values at $x_{sv}>0.5$ (Fig. 2).

$e_s$ changes more realistic of the atmosphere were also tested.  Results for 60-80% and 80-60% (Fig. A4) are similar

10  to those for 1-90% and 90-1% for their respective $D_i$ dependency; there is good model agreement, and across the $D_{p,t=0}$ and $D_{nv}^0$ range and for both dependencies $e$-folding time is less than 1 s.  Results for 10-20% and 20-10%, given in Fig. 4, also show agreement between models.  For both dependencies diffusion is much slower than in the 1-90% and 60-80% simulations, approaching 1 ky at low $D_{nv}^0$ and high $D_{p,t=0}$.  This shows that at low saturation ratios of semi-volatile component, gas-particle partitioning can be limited by condensed-phase diffusion in viscous particles.

15  $e$-folding times between models were also found to be in good agreement for these changes in $e_s$ when $D_{nv}^0$ was fixed at $1.0\times10^{-25}$ m$^2$s$^{-1}$ and $D_{sv}^0$ was varied between $1.0\times10^{-25}$ - $2\times10^{-9}$ m$^2$s$^{-1}$.  As discussed, the agreement between models in estimating $e$-folding times indicates that the estimated profiles of concentration with particle radius prior to the $e$-folding state are consistent between models because the underlying equations are the same.  By comparing concentration-radius profiles at various stages of diffusion we indeed found good model agreement across all cases.  In Fig. 5 we show the

20  example of the logarithmic dependence of $D_i$ on $x_{sv}$, an instantaneous change in saturation ratio of 1-90% and with $D_{nv}^0 = 1\times10^{-25}$ m$^2$s$^{-1}$ and $D_{sv}^0 = 2\times10^{-9}$ m$^2$s$^{-1}$.  At several times preceding and including $e$-folding time the concentration-radius profiles are in good agreement.

In the final part of this study the estimated temporal profile of particle radius was compared between the sigmoidal and logarithmic $D_i$ dependencies.  We have used the water activity and density dependence on sucrose weight fraction as

25  described in Zobrist et al. (2011) for the sucrose-water system in an attempt to replicate a non-ideal system.  The ETH model was employed, though the results above indicate that KM-GAP and Fi-PaD would produce identical profiles.  For the inference of $D_i$ dependency from radius measurements the signal to noise ratio is minimised by inducing a large change in radius relative to the measurement accuracy over a time-span that is large compared to the measurement frequency.

Taking the case of water as the semi-volatile component, from Fig. 3 it is clear that for certain values of $D_{nv}^0$ and

30  certain changes in $e_s$ attaining a large ratio of equilibrium time to measurement frequency may be difficult, even if the change in radius is large.  Indeed, the radius-time profiles in Figs. 6 and 7 for instantaneous changes in $e_s$ and a $D_{nv}^0 = 1\times10^{-25}$ m$^2$s$^{-1}$ confirm that for changes with a high final $e_s$, significant radius change is estimated to occur over less than 1 s, while the measurement frequency reported in the studies of Zobrist et al. (2011) and Lienhard et al. (2014) is approximately 15 s.

Simon O'Meara 15/3/16 12:38
Comment [10]: reviewer 2 comment 4: consistency in concentration-radius profile estimates presented

Simon O'Meara 15/3/16 12:42
Comment [11]: Reviewer 1 comment 5: more information about how non-ideality accounted for

Nevertheless, for the $e_s$ change of 1-90% in Fig. 5, there is a notable difference in the radius profiles between the dependencies. Despite having lower $D_i$ at low $x_{sv}$, the radius change from the sigmoidal dependence is more rapid than the logarithmic, indicating that the $D_i$ at higher $x_{sv}$ has a dominating effect on the profile. The inference of $D_i$ dependency using such a large change in $e_s$ is therefore poorly constrained for lower $x_{sv}$. For better constraint smaller changes in $e_s$ are

5    required, such as those used in Lienhard et al. (2014). An example of the radius profiles following incremental changes in $e_s$, $D_{nv}^0 = 1\text{x}10^{-25}$ m$^2$s$^{-1}$, $D_{sv}^0 = 2\text{x}10^{-9}$ m$^2$s$^{-1}$ and using both dependencies is shown in Fig. 7. This plot demonstrates the need for consideration of the time a given $e_s$ is maintained in measurement experiments, since the difference in the equilibrium timescales between the $e_s$ increments covers orders of magnitude. Indeed, over low changes in $e_s$ such as between 1-10%, equilibration time may be too long to be practical for gaining a useful measurement of radius change. It is worthwhile to

10    note that the rate of change of $e_s$ over an increment is preferably much greater than the rate of equilibration, as this provides the greatest potential for a clear signature of the $D_i$ dependence and therefore greatest constraint on inference.

**4 Discussion and Conclusion**

The results above show that despite variations in their numerical methods, all three Fickian-based diffusion models tested here: the ETH model, KM-GAP and Fi-PaD give good agreement of estimated $e$-folding timescales over a wide range of

15    changes to the saturation ratio of the semi-volatile component and over a wide range of differences in the self-diffusion coefficient of the semi-volatile and non-volatile components. Furthermore, there is good agreement between models when different dependencies of diffusion coefficient on composition are used. This result has not been reported before to our knowledge and verifies consistency between existing Fickian diffusion models. The maximum disagreement in $e$-folding times for results gained with satisfactory shell resolution is 10%, which is within the uncertainty generated by varying

20    degrees of numerical convergence and potential numerical diffusion. The consistency in modelled concentration-radius profiles at times preceding and including the $e$-folding state (Fig. 5) shows that if used for a polydisperse aerosol population, the models would give agreement in changes to the size distribution. In addition, if the diffusing component were reactive the rate of particle-phase reaction would depend on its concentration; therefore model agreement in concentration-radius profiles would give consistent reaction rates across the particle (which in turn could affect diffusion rate).

25      Using the three diffusion models as described above and with the spatial resolutions presented in the appendix, the ETH model takes approximately two orders of magnitude less computer time than Fi-PaD or KM-GAP for a given diffusion scenario. With the models giving consistent estimates of diffusion, the ETH model therefore appears to be favourable.

     The $e$-folding times given in Fig. 3 for changes in $e_s$ of 1-90% and 90-1%, and in Fig. A4 for changes of 60-80% and 80-60%, show that for a semi-volatile component with water-like (at room temperature) diffusivity, given a sufficiently

30    high starting/finishing $e_s$, attainment of the $e$-folding state is effectively instant compared to residence times in the atmosphere and chamber experiments. This is due to the plasticising effect of water (and applies to any semi-volatile component with a sufficiently high self-diffusion coefficient). At lower values of $e_s$ diffusion time can be much longer (Fig.

Simon O'Meara 15/3/16 12:50

**Comment [12]:** reviewer 2 comment 4: discussion on the implications of concentration-radius profile agreement

Simon O'Meara 3/4/16 08:14

**Comment [13]:** reviewer 2 comment 5) - the order of magnitude for computer time taken by each model is compared for two extreme cases.

[revised manuscript text omitted]